# Low-Rank Regression with Tensor Responses

**Guillaume Rabusseau and Hachem Kadri**
Aix Marseille Univ, CNRS, LIF, Marseille, France
`{firstname.lastname}@lif.univ-mrs.fr`

## Abstract

This paper proposes an efficient algorithm (HOLRR) to handle regression tasks where the outputs have a tensor structure. We formulate the regression problem as the minimization of a least square criterion under a multilinear rank constraint, a difficult non convex problem. HOLRR computes efficiently an approximate solution of this problem, with solid theoretical guarantees. A kernel extension is also presented. Experiments on synthetic and real data show that HOLRR computes accurate solutions while being computationally very competitive.

## 1 Introduction

Recently, there has been an increasing interest in adapting machine learning and statistical methods to tensors. Data with a natural tensor structure are encountered in many scientific areas including neuroimaging [30], signal processing [4], spatio-temporal analysis [2] and computer vision [16]. Extending multivariate regression methods to tensors is one of the challenging task in this area. Most existing works extend linear models to the multilinear setting and focus on the tensor structure of the input data (e.g. [24]). Little has been done however to investigate learning methods for *tensor-structured output data*.

We consider a multilinear regression task where outputs are tensors; such a setting can occur in the context of e.g. spatio-temporal forecasting or image reconstruction. In order to leverage the tensor structure of the output data, we formulate the problem as the minimization of a least squares criterion subject to a *multilinear rank* constraint on the regression tensor. The rank constraint enforces the model to capture low-rank structure in the outputs and to explain dependencies between inputs and outputs in a low-dimensional multilinear subspace.

Unlike previous work (e.g. [22, 24, 27]) we do not rely on a convex relaxation of this difficult non-convex optimization problem. Instead we show that it is equivalent to a multilinear subspace identification problem for which we design a fast and efficient approximation algorithm (HOLRR), along with a kernelized version which extends our approach to the nonlinear setting (Section 3). Our theoretical analysis shows that HOLRR provides good approximation guarantees. Furthermore, we derive a generalization bound for the class of tensor-valued regression functions with bounded multilinear rank (Section 3.3). Experiments on synthetic and real data are presented to validate our theoretical findings and show that HOLRR computes accurate solutions while being computationally very competitive (Section 4).

Proofs of all results stated in the paper can be found in supplementary material A.

**Related work.**    The problem we consider is a generalization of the reduced-rank regression problem (Section 2.2) to tensor structured responses. Reduced-rank regression has its roots in statistics [10] but it has also been investigated by the neural network community [3]; non-parametric extensions of this method have been proposed in [18] and [6]. In the context of multi-task learning, a linear model using a tensor-rank penalization of a least squares criterion has been proposed in [22] to take into account the multi-modal interactions between

tasks. They propose an approach relying on a convex relaxation of the multlinear rank constraint using the trace norms of the matricizations, and a non-convex approach based on alternating minimization. Nonparametric low-rank estimation strategies in reproducing kernel Hilbert spaces (RKHS) based on a multilinear spectral regularization have been proposed in [23, 24]. Their method is based on estimating the regression function in the tensor product of RKHSs and is naturally adapted for tensor covariates. A greedy algorithm to solve a low-rank tensor learning problem has been proposed in [2] in the context of multivariate spatio-temporal data analysis. The linear model they assume is different from the one we propose and is specifically designed for spatio-temporal data. A higher-order extension of partial least squares (HOPLS) has been proposed in [28] along with a kernel extension in [29]. While HOPLS has the advantage of taking the tensor structure of the input into account, the questions of approximation and generalization guarantees were not addressed in [28]. The generalization bound we provide is inspired from works on matrix and tensor completion [25, 19].

## 2 Preliminaries

We begin by introducing some notations. For any integer $k$ we use $[k]$ to denote the set of integers from 1 to $k$. We use lower case bold letters for vectors (e.g. $\mathbf{v} \in \mathbb{R}^{d_1}$), upper case bold letters for matrices (e.g. $\mathbf{M} \in \mathbb{R}^{d_1 \times d_2}$) and bold calligraphic letters for higher order tensors (e.g. $\boldsymbol{\mathcal{T}} \in \mathbb{R}^{d_1 \times d_2 \times d_3}$). The identity matrix will be written as $\mathbf{I}$. The $i$th row (resp. column) of a matrix $\mathbf{M}$ will be denoted by $\mathbf{M}_{i,:}$ (resp. $\mathbf{M}_{:,i}$). This notation is extended to slices of a tensor in the straightforward way. If $\mathbf{v} \in \mathbb{R}^{d_1}$ and $\mathbf{v}' \in \mathbb{R}^{d_2}$, we use $\mathbf{v} \otimes \mathbf{v}' \in \mathbb{R}^{d_1 \cdot d_2}$ to denote the Kronecker product between vectors, and its straightforward extension to matrices and tensors. Given a matrix $\mathbf{M} \in \mathbb{R}^{d_1 \times d_2}$, we use $\mathrm{vec}(\mathbf{M}) \in \mathbb{R}^{d_1 \cdot d_2}$ to denote the column vector obtained by concatenating the columns of $\mathbf{M}$.

### 2.1 Tensors and Tucker Decomposition

We first recall basic definitions of tensor algebra; more details can be found in [13]. A *tensor* $\boldsymbol{\mathcal{T}} \in \mathbb{R}^{d_1 \times \cdots \times d_p}$ can simply be seen as a multidimensional array $(\boldsymbol{\mathcal{T}}_{i_1, \cdots, i_p} : i_n \in [d_n], n \in [p])$. The *mode-n fibers* of $\boldsymbol{\mathcal{T}}$ are the vectors obtained by fixing all indices except the $n$th one, e.g. $\boldsymbol{\mathcal{T}}_{:, i_2, \cdots, i_p} \in \mathbb{R}^{d_1}$. The *nth mode matricization* of $\boldsymbol{\mathcal{T}}$ is the matrix having the mode-$n$ fibers of $\boldsymbol{\mathcal{T}}$ for columns and is denoted by $\mathbf{T}_{(n)} \in \mathbb{R}^{d_n \times d_1 \cdots d_{n-1} d_{n+1} \cdots d_p}$. The vectorization of a tensor is defined by $\mathrm{vec}(\boldsymbol{\mathcal{T}}) = \mathrm{vec}(\mathbf{T}_{(1)})$. The *inner product* between two tensors $\boldsymbol{\mathcal{S}}$ and $\boldsymbol{\mathcal{T}}$ (of the same size) is defined by $\langle \boldsymbol{\mathcal{S}}, \boldsymbol{\mathcal{T}} \rangle = \langle \mathrm{vec}(\boldsymbol{\mathcal{S}}), \mathrm{vec}(\boldsymbol{\mathcal{T}}) \rangle$ and the Frobenius norm is defined by $\|\boldsymbol{\mathcal{T}}\|_F^2 = \langle \boldsymbol{\mathcal{T}}, \boldsymbol{\mathcal{T}} \rangle$. In the following $\boldsymbol{\mathcal{T}}$ always denotes a tensor of size $d_1 \times \cdots \times d_p$.

The *mode-n matrix product* of the tensor $\boldsymbol{\mathcal{T}}$ and a matrix $\mathbf{X} \in \mathbb{R}^{m \times d_n}$ is a tensor denoted by $\boldsymbol{\mathcal{T}} \times_n \mathbf{X}$. It is of size $d_1 \times \cdots \times d_{n-1} \times m \times d_{n+1} \times \cdots \times d_p$ and is defined by the relation $\boldsymbol{\mathcal{Y}} = \boldsymbol{\mathcal{T}} \times_n \mathbf{X} \Leftrightarrow \mathbf{Y}_{(n)} = \mathbf{X}\mathbf{T}_{(n)}$. The *mode-n vector product* of the tensor $\boldsymbol{\mathcal{T}}$ and a vector $\mathbf{v} \in \mathbb{R}^{d_n}$ is a tensor defined by $\boldsymbol{\mathcal{T}} \bullet_n \mathbf{v} = \boldsymbol{\mathcal{T}} \times_n \mathbf{v}^\top \in \mathbb{R}^{d_1 \times \cdots \times d_{n-1} \times d_{n+1} \times \cdots \times d_p}$. The *mode-n rank* of $\boldsymbol{\mathcal{T}}$ is the dimension of the space spanned by its mode-$n$ fibers, that is $\mathrm{rank}_n(\boldsymbol{\mathcal{T}}) = \mathrm{rank}(\mathbf{T}_{(n)})$. The *multilinear rank* of $\boldsymbol{\mathcal{T}}$, denoted by $\mathrm{rank}(\boldsymbol{\mathcal{T}})$, is the tuple of mode-$n$ ranks of $\boldsymbol{\mathcal{T}}$: $rank(\boldsymbol{\mathcal{T}}) = (R_1, \cdots, R_p)$ where $R_n = \mathrm{rank}_n(\boldsymbol{\mathcal{T}})$ for $n \in [p]$. We will write $\mathrm{rank}(\boldsymbol{\mathcal{T}}) \leq (S_1, \cdots, S_p)$ whenever $\mathrm{rank}_1(\boldsymbol{\mathcal{T}}) \leq S_1, \mathrm{rank}_2(\boldsymbol{\mathcal{T}}) \leq S_2, \cdots, \mathrm{rank}_p(\boldsymbol{\mathcal{T}}) \leq S_p$.

The *Tucker decomposition* decomposes a tensor $\boldsymbol{\mathcal{T}}$ into a core tensor $\boldsymbol{\mathcal{G}}$ transformed by an orthogonal matrix along each mode: (i) $\boldsymbol{\mathcal{T}} = \boldsymbol{\mathcal{G}} \times_1 \mathbf{U}_1 \times_2 \mathbf{U}_2 \times_3 \cdots \times_p \mathbf{U}_p$, where $\boldsymbol{\mathcal{G}} \in \mathbb{R}^{R_1 \times R_2 \times \cdots \times R_p}$, $\mathbf{U}_i \in \mathbb{R}^{d_i \times R_i}$ and $\mathbf{U}_i^\top \mathbf{U}_i = \mathbf{I}$ for all $i \in [p]$. The number of parameters involved in a Tucker decomposition can be considerably smaller than $d_1 d_2 \cdots d_p$. We have the following identities when matricizing and vectorizing a Tucker decomposition: $\mathbf{T}_{(n)} = \mathbf{U}_n \mathbf{G}_{(n)} (\mathbf{U}_p \otimes \cdots \otimes \mathbf{U}_{n+1} \otimes \mathbf{U}_{n-1} \otimes \cdots \otimes \mathbf{U}_1)^\top$ and $\mathrm{vec}(\boldsymbol{\mathcal{T}}) = (\mathbf{U}_p \otimes \mathbf{U}_{p-1} \otimes \cdots \otimes \mathbf{U}_1)\mathrm{vec}(\boldsymbol{\mathcal{G}})$.

It is well known that $\boldsymbol{\mathcal{T}}$ admits the Tucker decomposition (i) iff $\mathrm{rank}(\boldsymbol{\mathcal{T}}) \leq (R_1, \cdots, R_p)$ (see e.g. [13]). Finding an exact Tucker decomposition can be done using the higher-order SVD algorithm (HOSVD) introduced by [5]. Although finding the best approximation of

multilinear rank $(R_1, \cdots, R_p)$ of a tensor $\mathcal{T}$ is a difficult problem, the truncated HOSVD algorithm provides good approximation guarantees and often performs well in practice.

## 2.2 Low-Rank Regression

Multivariate regression is the task of recovering a function $f : \mathbb{R}^d \to \mathbb{R}^p$ from a set of input-output pairs $\{(\mathbf{x}^{(n)}, \mathbf{y}^{(n)})\}_{n=1}^N$ sampled from the model with an additive noise $\mathbf{y} = f(\mathbf{x}) + \boldsymbol{\varepsilon}$, where $\boldsymbol{\varepsilon}$ is the error term. To solve this problem, the *ordinary least squares* (OLS) approach assumes a linear dependence between input and output data and boils down to finding a matrix $\mathbf{W} \in \mathbb{R}^{d \times p}$ that minimizes the squared error $\|\mathbf{XW} - \mathbf{Y}\|_F^2$, where $\mathbf{X} \in \mathbb{R}^{N \times d}$ and $\mathbf{Y} \in \mathbb{R}^{N \times p}$ denote the input and the output matrices. To prevent overfitting and to avoid numerical instabilities a ridge regularization term (i.e. $\gamma\|\mathbf{W}\|_F^2$) is often added to the objective function, leading to the *regularized least squares* (RLS) method. It is easy to see that the OLS/RLS approach in the multivariate setting is equivalent to performing $p$ linear regressions for each scalar output $\{\mathbf{y}_j\}_{j=1}^p$ independently. Thus it performs poorly when the outputs are correlated and the true dimension of the response is less than $p$. *Low-rank regression* (or reduced-rank regression) addresses this issue by solving the rank penalized problem $\min_{\mathbf{W} \in \mathbb{R}^{d \times p}} \|\mathbf{XW} - \mathbf{Y}\|_F^2 + \gamma\|\mathbf{W}\|_F^2$ s.t. $\text{rank}(\mathbf{W}) \le R$ for a given integer $R$. The rank constraint was first proposed in [1], whereas the term *reduced-rank regression* was introduced in [10]. Adding a ridge regularization was proposed in [18]. In the rest of the paper we will refer to this approach as low-rank regression (LRR). For more description and discussion of reduced-rank regression, we refer the reader to the books [21] and [11].

# 3 Low-Rank Regression for Tensor-Valued Functions

## 3.1 Problem Formulation

We consider a multivariate regression task where the input is a vector and the response has a tensor structure. Let $f : \mathbb{R}^{d_0} \to \mathbb{R}^{d_1 \times d_2 \times \cdots \times d_p}$ be the function we want to learn from a sample of input-output data $\{(\mathbf{x}^{(n)}, \boldsymbol{\mathcal{Y}}^{(n)})\}_{n=1}^N$ drawn from the model $\boldsymbol{\mathcal{Y}} = f(\mathbf{x}) + \boldsymbol{\mathcal{E}}$, where $\boldsymbol{\mathcal{E}}$ is an error term. We assume that $f$ is linear, that is $f(\mathbf{x}) = \boldsymbol{\mathcal{W}} \bullet_1 \mathbf{x}$ for some regression tensor $\boldsymbol{\mathcal{W}} \in \mathbb{R}^{d_0 \times d_1 \times \cdots \times d_p}$. The vectorization of this relation leads to $\text{vec}(f(\mathbf{x})) = \mathbf{W}_{(1)}^\top \mathbf{x}$ showing that this model is equivalent to the standard multivariate linear model. One way to tackle this regression task would be to vectorize each output sample and to perform a standard low-rank regression on the data $\{(\mathbf{x}^{(n)}, \text{vec}(\boldsymbol{\mathcal{Y}}^{(n)}))\}_{n=1}^N \subset \mathbb{R}^{d_0} \times \mathbb{R}^{d_1 \cdots d_p}$. A major drawback of this approach is that the tensor structure of the output is lost in the vectorization step. The low-rank model tries to capture linear dependencies between components of the output but it ignores *higher level dependencies* that could be present in a tensor-structured output. For illustration, suppose the output is a matrix encoding the samples of $d_1$ continuous variables at $d_2$ different time steps, one could expect structural relations between the $d_1$ time series, e.g. linear dependencies between the rows of the output matrix.

**Low-rank regression for tensor responses.** To overcome the limitation described above we propose an extension of the low-rank regression method for tensor-structured responses by enforcing low multilinear rank of the regression tensor $\boldsymbol{\mathcal{W}}$. Let $\{(\mathbf{x}^{(n)}, \boldsymbol{\mathcal{Y}}^{(n)})\}_{n=1}^N \subset \mathbb{R}^{d_0} \times \mathbb{R}^{d_1 \times d_2 \times \cdots \times d_p}$ be a training sample of input/output data drawn from the model $f(\mathbf{x}) = \boldsymbol{\mathcal{W}} \bullet_1 \mathbf{x} + \boldsymbol{\mathcal{E}}$ where $\boldsymbol{\mathcal{W}}$ is assumed of low multilinear rank. Considering the framework of empirical risk minimization, we want to find a low-rank regression tensor $\boldsymbol{\mathcal{W}}$ minimizing the loss on the training data. To avoid numerical instabilities and to prevent overfitting we add a ridge regularization to the objective function, leading to the minimization of $\sum_{n=1}^N \ell(\boldsymbol{\mathcal{W}} \bullet_1 \mathbf{x}^{(n)}, \boldsymbol{\mathcal{Y}}^{(n)}) + \gamma\|\boldsymbol{\mathcal{W}}\|_F^2$ w.r.t. the regression tensor $\boldsymbol{\mathcal{W}}$ subject to the constraint $\text{rank}(\boldsymbol{\mathcal{W}}) \le (R_0, R_1, \cdots, R_p)$ for some given integers $R_0, R_1, \cdots, R_p$ and where $\ell$ is a loss function. In this paper, we consider the squared error loss between tensors defined by $\mathcal{L}(\boldsymbol{\mathcal{T}}, \hat{\boldsymbol{\mathcal{T}}}) = \|\boldsymbol{\mathcal{T}} - \hat{\boldsymbol{\mathcal{T}}}\|_F^2$. Using this loss we can rewrite the minimization problem as

$$\min_{\boldsymbol{\mathcal{W}} \in \mathbb{R}^{d_0 \times d_1 \times \cdots \times d_p}} \|\boldsymbol{\mathcal{W}} \times_1 \mathbf{X} - \boldsymbol{\mathcal{Y}}\|_F^2 + \gamma\|\boldsymbol{\mathcal{W}}\|_F^2 \quad \text{s.t. } \text{rank}(\boldsymbol{\mathcal{W}}) \le (R_0, R_1, \cdots, R_p), \quad (1)$$

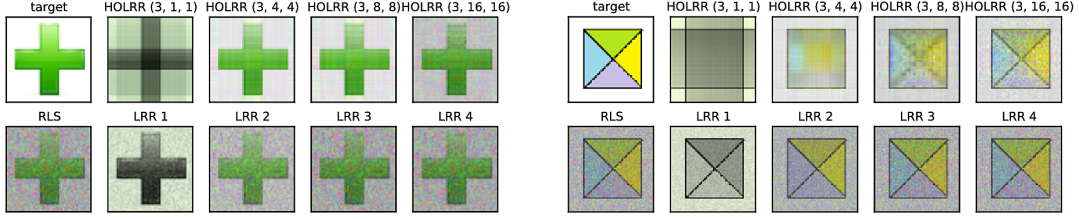

Figure 1: Image reconstruction from noisy measurements: $\boldsymbol{\mathcal{Y}} = \boldsymbol{\mathcal{W}} \bullet_1 \mathbf{x} + \boldsymbol{\mathcal{E}}$ where $\boldsymbol{\mathcal{W}}$ is a color image (RGB). Each image is labeled with the algorithm and the rank parameter.

where the input matrix $\mathbf{X} \in \mathbb{R}^{N \times d_0}$ and the output tensor $\boldsymbol{\mathcal{Y}} \in \mathbb{R}^{N \times d_1 \times \cdots \times d_p}$ are defined by $\mathbf{X}_{n,:} = (\mathbf{x}^{(n)})^\top$, $\boldsymbol{\mathcal{Y}}_{n,:,\cdots,:} = \boldsymbol{\mathcal{Y}}^{(n)}$ for $n = 1, \cdots, N$ ($\boldsymbol{\mathcal{Y}}$ is the tensor obtained by stacking the output tensors along the first mode).

**Low-rank regression function.** Let $\boldsymbol{\mathcal{W}}^*$ be a solution of problem (1), it follows from the multilinear rank constraint that $\boldsymbol{\mathcal{W}}^* = \boldsymbol{\mathcal{G}} \times_1 \mathbf{U}_0 \times_2 \cdots \times_{p+1} \mathbf{U}_p$ for some core tensor $\boldsymbol{\mathcal{G}} \in \mathbb{R}^{R_0 \times \cdots \times R_p}$ and orthogonal matrices $\mathbf{U}_i \in \mathbb{R}^{d_i \times R_i}$ for $0 \leq i \leq p$. The regression function $f^* : \mathbf{x} \mapsto \boldsymbol{\mathcal{W}}^* \bullet_1 \mathbf{x}$ can thus be written as $f^* : \mathbf{x} \mapsto \boldsymbol{\mathcal{G}} \times_1 \mathbf{x}^\top \mathbf{U}_0 \times_2 \cdots \times_{p+1} \mathbf{U}_p$.

This implies several interesting properties. First, for any $\mathbf{x} \in \mathbb{R}^{d_0}$ we have $f^*(\mathbf{x}) = \boldsymbol{\mathcal{T}}_\mathbf{x} \times_1 \mathbf{U}_1 \times_2 \cdots \times_p \mathbf{U}_p$ with $\boldsymbol{\mathcal{T}}_\mathbf{x} = \boldsymbol{\mathcal{G}} \bullet_1 \mathbf{U}_0^\top \mathbf{x}$, which implies $\text{rank}(f^*(\mathbf{x})) \leq (R_1, \cdots, R_p)$, that is the image of $f^*$ is a set of tensors with low multilinear rank. Second, the relation between $\mathbf{x}$ and $\boldsymbol{\mathcal{Y}} = f^*(\mathbf{x})$ is explained in a low dimensional subspace of size $R_0 \times R_1 \times \cdots \times R_p$. Indeed one can decompose the mapping $f^*$ into the following steps: (i) project $\mathbf{x}$ in $\mathbb{R}^{R_0}$ as $\bar{\mathbf{x}} = \mathbf{U}_0^\top \mathbf{x}$, (ii) perform a low-dimensional mapping $\bar{\boldsymbol{\mathcal{Y}}} = \boldsymbol{\mathcal{G}} \bullet_1 \bar{\mathbf{x}}$, (iii) project back into the output space to get $\boldsymbol{\mathcal{Y}} = \bar{\boldsymbol{\mathcal{Y}}} \times_1 \mathbf{U}_1 \times_2 \cdots \times_p \mathbf{U}_p$.

To give an illustrative intuition on the differences between matrix and multilinear rank regularization we present a simple experiment[1] in Figure 1. We generate data from the model $\boldsymbol{\mathcal{Y}} = \boldsymbol{\mathcal{W}} \bullet_1 \mathbf{x} + \boldsymbol{\mathcal{E}}$ where the tensor $\boldsymbol{\mathcal{W}} \in \mathbb{R}^{3 \times m \times n}$ is a color image of size $m \times n$ encoded with three color channels RGB. The components of both $\mathbf{x}$ and $\boldsymbol{\mathcal{E}}$ are drawn from $\mathcal{N}(0,1)$. This experiment allows us to visualize the tensors returned by RLS, LRR and our method HOLRR that enforces low multilinear rank of the regression function. First, this shows that the function learned by vectorizing the outputs and performing LRR does not enforce any low-rank structure. This is well illustrated in (Figure 1) where the regression tensors returned by HOLRR-(3,1,1) are clearly of low-rank while the ones returned by LRR-1 are not. This also shows that taking into account the low-rank structure of the model allows one to better eliminate the noise when the true regression tensor is of low rank (Figure 1, left). However if the ground truth model does not have a low-rank structure, enforcing low mutlilinear rank leads to underfitting for low values of the rank parameter (Figure 1, right).

## 3.2 Higher-Order Low-Rank Regression and its Kernel Extension

We now propose an efficient algorithm to tackle problem (1). We first show that the ridge regularization term in (1) can be incorporated in the data fitting term. Let $\tilde{\mathbf{X}} \in \mathbb{R}^{(N+d_0) \times d_0}$ and $\tilde{\boldsymbol{\mathcal{Y}}} \in \mathbb{R}^{(N+d_0) \times d_1 \times \cdots \times d_p}$ be defined by $\tilde{\mathbf{X}}^\top = (\mathbf{X} \mid \gamma \mathbf{I})^\top$ and $\tilde{\mathbf{Y}}_{(1)}^\top = (\mathbf{Y}_{(1)} \mid \mathbf{0})^\top$. It is easy to check that the objective function in (1) is equal to $\|\boldsymbol{\mathcal{W}} \times_1 \tilde{\mathbf{X}} - \tilde{\boldsymbol{\mathcal{Y}}}\|_F^2$. Minimization problem (1) is then equivalent to

$$\min_{\substack{\boldsymbol{\mathcal{G}} \in \mathbb{R}^{R_0 \times R_1 \times \cdots \times R_p}, \\ \mathbf{U}_i \in \mathbb{R}^{d_i \times R_i} \text{ for } 0 \leq i \leq p}} \|\boldsymbol{\mathcal{W}} \times_1 \tilde{\mathbf{X}} - \tilde{\boldsymbol{\mathcal{Y}}}\|_F^2 \quad \text{s.t. } \boldsymbol{\mathcal{W}} = \boldsymbol{\mathcal{G}} \times_1 \mathbf{U}_0 \cdots \times_{p+1} \mathbf{U}_p, \mathbf{U}_i^\top \mathbf{U}_i = \mathbf{I} \text{ for all } i. \quad (2)$$

We now show that this minimization problem can be reduced to finding $p + 1$ projection matrices onto subspaces of dimension $R_0, R_1, \cdots, R_p$. We start by showing that the core tensor $\boldsymbol{\mathcal{G}}$ solution of (2) is determined by the factor matrices $\mathbf{U}_0, \cdots, \mathbf{U}_p$.

**Theorem 1.** *For given orthogonal matrices* $\mathbf{U}_0, \cdots, \mathbf{U}_p$ *the tensor* $\boldsymbol{\mathcal{G}}$ *that minimizes (2) is given by* $\boldsymbol{\mathcal{G}} = \tilde{\boldsymbol{\mathcal{Y}}} \times_1 (\mathbf{U}_0^\top \tilde{\mathbf{X}}^\top \tilde{\mathbf{X}} \mathbf{U}_0)^{-1} \mathbf{U}_0^\top \tilde{\mathbf{X}}^\top \times_2 \mathbf{U}_1^\top \times_3 \cdots \times_{p+1} \mathbf{U}_p^\top.$

It follows from Theorem 1 that problem (1) can be written as

$$\min_{\mathbf{U}_i \in \mathbb{R}^{d_i \times R_i}, 0 \le i \le p} \| \tilde{\boldsymbol{\mathcal{Y}}} \times_1 \boldsymbol{\Pi}_0 \times_2 \cdots \times_{p+1} \boldsymbol{\Pi}_p - \tilde{\boldsymbol{\mathcal{Y}}} \|_F^2 \qquad (3)$$

subject to $\mathbf{U}_i^\top \mathbf{U}_i = \mathbf{I}$ for all $i$, $\boldsymbol{\Pi}_0 = \tilde{\mathbf{X}} \mathbf{U}_0 \left( \mathbf{U}_0^\top \tilde{\mathbf{X}}^\top \tilde{\mathbf{X}} \mathbf{U}_0 \right)^{-1} \mathbf{U}_0^\top \tilde{\mathbf{X}}^T$, $\boldsymbol{\Pi}_i = \mathbf{U}_i \mathbf{U}_i^\top$ for $i \ge 1$. Note that $\boldsymbol{\Pi}_0$ is the orthogonal projection onto the space spanned by the columns of $\tilde{\mathbf{X}} \mathbf{U}_0$ and $\boldsymbol{\Pi}_i$ is the orthogonal projection onto the column space of $\mathbf{U}_i$ for $i \ge 1$. Hence solving problem (1) is equivalent to finding $p+1$ low-dimensional subspaces $U_0, \cdots, U_p$ such that projecting $\tilde{\boldsymbol{\mathcal{Y}}}$ onto the spaces $\tilde{\mathbf{X}} U_0, U_1, \cdots, U_p$ along the corresponding modes is close to $\tilde{\boldsymbol{\mathcal{Y}}}$.

**HOLRR algorithm.** Since solving problem (3) for the $p+1$ projections simultaneously is a difficult non-convex optimization problem we propose to solve it independently for each projection. This approach has the benefits of both being computationally efficient and providing good theoretical approximation guarantees (see Theorem 2). The following proposition gives the analytic solutions of (3) when each projection is considered independently.

**Proposition 1.** *For* $0 \le i \le p$, *using the definition of* $\boldsymbol{\Pi}_i$ *in (3), the optimal solution of* $\min_{\mathbf{U}_i \in \mathbb{R}^{d_i \times R_i}} \| \tilde{\boldsymbol{\mathcal{Y}}} \times_{i+1} \boldsymbol{\Pi}_i - \tilde{\boldsymbol{\mathcal{Y}}} \|_F^2$ *s.t.* $\mathbf{U}_i^\top \mathbf{U}_i = \mathbf{I}$ *is given by the top* $R_i$ *eigenvectors of* $(\tilde{\mathbf{X}}^\top \tilde{\mathbf{X}})^{-1} \tilde{\mathbf{X}}^\top \tilde{\mathbf{Y}}_{(1)} \tilde{\mathbf{Y}}_{(1)}^\top \tilde{\mathbf{X}}$ *if* $i = 0$ *and* $\tilde{\mathbf{Y}}_{(i+1)} \tilde{\mathbf{Y}}_{(i+1)}^\top$ *otherwise.*

The results from Theorem 1 and Proposition 1 can be rewritten in terms of the original input matrix $\mathbf{X}$ and output tensor $\boldsymbol{\mathcal{Y}}$ using the identities $\tilde{\mathbf{X}}^\top \tilde{\mathbf{X}} = \mathbf{X}^\top \mathbf{X} + \gamma \mathbf{I}$, $\tilde{\boldsymbol{\mathcal{Y}}} \times_1 \tilde{\mathbf{X}}^\top = \boldsymbol{\mathcal{Y}} \times_1 \mathbf{X}^\top$ and $\tilde{\mathbf{Y}}_{(i)} \tilde{\mathbf{Y}}_{(i)}^\top = \mathbf{Y}_{(i)} \mathbf{Y}_{(i)}^\top$ for any $i \ge 1$. The overall Higher-Order Low-Rank Regression procedure (HOLRR) is summarized in Algorithm 1. Note that the Tucker decomposition of the solution returned by HOLRR could be a good initialization point for an Alternative Least Square method. However, studying the theoretical and experimental properties of this approach is beyond the scope of this paper and is left for future work.

**HOLRR Kernel Extension** We now design a kernelized version of the HOLRR algorithm by analyzing how it would be instantiated in a feature space. We show that all the steps involved can be performed using the Gram matrix of the input data without having to explicitly compute the feature map. Let $\phi : \mathbb{R}^{d_0} \to \mathbb{R}^L$ be a feature map and let $\boldsymbol{\Phi} \in \mathbb{R}^{N \times L}$ be the matrix with rows $\phi(\mathbf{x}^{(n)})^\top$ for $n \in [N]$. The higher-order low-rank regression problem in the feature space boils down to the minimization problem

$$\min_{\boldsymbol{\mathcal{W}} \in \mathbb{R}^{L \times d_1 \times \cdots \times d_p}} \| \boldsymbol{\mathcal{W}} \times_1 \boldsymbol{\Phi} - \boldsymbol{\mathcal{Y}} \|_F^2 + \gamma \| \boldsymbol{\mathcal{W}} \|_F^2 \qquad \text{s.t. } \text{rank}(\boldsymbol{\mathcal{W}}) \le (R_0, R_1, \cdots, R_p) \ . \quad (4)$$

Following the HOLRR algorithm, one needs to compute the top $R_0$ eigenvectors of the $L \times L$ matrix $(\boldsymbol{\Phi}^\top \boldsymbol{\Phi} + \gamma \mathbf{I})^{-1} \boldsymbol{\Phi}^\top \mathbf{Y}_{(1)} \mathbf{Y}_{(1)}^\top \boldsymbol{\Phi}$. The following proposition shows that this can be done using the Gram matrix $\mathbf{K} = \boldsymbol{\Phi} \boldsymbol{\Phi}^\top$ without explicitly knowing the feature map $\phi$.

**Proposition 2.** *If* $\boldsymbol{\alpha} \in \mathbb{R}^N$ *is an eigenvector with eigenvalue* $\lambda$ *of the matrix* $(\mathbf{K} + \gamma \mathbf{I})^{-1} \mathbf{Y}_{(1)} \mathbf{Y}_{(1)}^\top \mathbf{K}$, *then* $\mathbf{v} = \boldsymbol{\Phi}^\top \boldsymbol{\alpha} \in \mathbb{R}^L$ *is an eigenvector with eigenvalue* $\lambda$ *of the matrix* $(\boldsymbol{\Phi}^\top \boldsymbol{\Phi} + \gamma \mathbf{I})^{-1} \boldsymbol{\Phi}^\top \mathbf{Y}_{(1)} \mathbf{Y}_{(1)}^\top \boldsymbol{\Phi}$.

Let $\mathbf{A}$ be the top $R_0$ eigenvectors of the matrix $(\mathbf{K} + \gamma \mathbf{I})^{-1} \mathbf{Y}_{(1)} \mathbf{Y}_{(1)}^\top \mathbf{K}$. When working with the feature map $\phi$, it follows from the previous proposition that line 1 in Algorithm 1 is equivalent to choosing $\mathbf{U}_0 = \boldsymbol{\Phi}^\top \mathbf{A} \in \mathbb{R}^{L \times R_0}$, while the updates in line 3 stay the same. The regression tensor $\boldsymbol{\mathcal{W}} \in \mathbb{R}^{L \times d_1 \times \cdots \times d_p}$ returned by this algorithm is then equal to $\boldsymbol{\mathcal{W}} = \boldsymbol{\mathcal{Y}} \times_1 \mathbf{P} \times_2 \mathbf{U}_1 \mathbf{U}_1^\top \times_2 \cdots \times_{p+1} \mathbf{U}_p \mathbf{U}_p^\top$, where $\mathbf{P} = \boldsymbol{\Phi}^\top \mathbf{A} \left( \mathbf{A}^\top \boldsymbol{\Phi} (\boldsymbol{\Phi}^\top \boldsymbol{\Phi} + \gamma \mathbf{I}) \boldsymbol{\Phi}^\top \mathbf{A} \right)^{-1} \mathbf{A}^\top \boldsymbol{\Phi} \boldsymbol{\Phi}^\top$. It is easy to check that $\mathbf{P}$ can be rewritten as $\mathbf{P} = \boldsymbol{\Phi}^\top \mathbf{A} \left( \mathbf{A}^\top \mathbf{K} (\mathbf{K} + \gamma \mathbf{I}) \mathbf{A} \right)^{-1} \mathbf{A}^\top \mathbf{K}$.

Suppose now that the feature map $\phi$ is induced by a kernel $k : \mathbb{R}^{d_0} \times \mathbb{R}^{d_0} \to \mathbb{R}$. The prediction for an input vector $\mathbf{x}$ is then given by $\boldsymbol{\mathcal{W}} \bullet_1 \mathbf{x} = \boldsymbol{\mathcal{C}} \bullet_1 \mathbf{k}_\mathbf{x}$ where the $n$th component

| **Algorithm 1** `HOLRR` | **Algorithm 2** `Kernelized HOLRR` |
|---|---|
| **Input:** $\mathbf{X} \in \mathbb{R}^{N \times d_0}$, $\boldsymbol{\mathcal{Y}} \in \mathbb{R}^{N \times d_1 \times \cdots \times d_p}$, rank $(R_0, R_1, \cdots, R_p)$ and regularization parameter $\gamma$. | **Input:** Gram matrix $\mathbf{K} \in \mathbb{R}^{N \times N}$, $\boldsymbol{\mathcal{Y}} \in \mathbb{R}^{N \times d_1 \times \cdots \times d_p}$, rank $(R_0, R_1, \cdots, R_p)$ and regularization parameter $\gamma$. |
| 1: $\mathbf{U}_0 \leftarrow$ top $R_0$ eigenvectors of $(\mathbf{X}^\top \mathbf{X} + \gamma \mathbf{I})^{-1} \mathbf{X}^\top \mathbf{Y}_{(1)} \mathbf{Y}_{(1)}^\top \mathbf{X}$ | 1: $\mathbf{A} \leftarrow$ top $R_0$ eigenvectors of $(\mathbf{K} + \gamma \mathbf{I})^{-1} \mathbf{Y}_{(1)} \mathbf{Y}_{(1)}^\top \mathbf{K}$ |
| 2: **for** $i = 1$ **to** $p$ **do** | 2: **for** $i = 1$ **to** $p$ **do** |
| 3:   $\mathbf{U}_i \leftarrow$ top $R_i$ eigenvec. of $\mathbf{Y}_{(i+1)} \mathbf{Y}_{(i+1)}^\top$ | 3:   $\mathbf{U}_i \leftarrow$ top $R_i$ eigenvec. of $\mathbf{Y}_{(i+1)} \mathbf{Y}_{(i+1)}^\top$ |
| 4: **end for** | 4: **end for** |
| 5: $\mathbf{M} = \left(\mathbf{U}_0^\top (\mathbf{X}^\top \mathbf{X} + \gamma \mathbf{I}) \mathbf{U}_0\right)^{-1} \mathbf{U}_0^\top \mathbf{X}^\top$ | 5: $\mathbf{M} \leftarrow \left(\mathbf{A}^\top \mathbf{K}(\mathbf{K} + \gamma \mathbf{I})\mathbf{A}\right)^{-1} \mathbf{A}^\top \mathbf{K}$ |
| 6: $\boldsymbol{\mathcal{G}} \leftarrow \boldsymbol{\mathcal{Y}} \times_1 \mathbf{M} \times_2 \mathbf{U}_1^\top \times_3 \cdots \times_{p+1} \mathbf{U}_p^\top$ | 6: $\boldsymbol{\mathcal{G}} \leftarrow \boldsymbol{\mathcal{Y}} \times_1 \mathbf{M} \times_2 \mathbf{U}_1^\top \times_3 \cdots \times_{p+1} \mathbf{U}_p^\top$ |
| 7: **return** $\boldsymbol{\mathcal{G}} \times_1 \mathbf{U}_0 \times_2 \cdots \times_{p+1} \mathbf{U}_p$ | 7: **return** $\boldsymbol{\mathcal{C}} = \boldsymbol{\mathcal{G}} \times_1 \mathbf{A} \times_2 \mathbf{U}_1 \times_3 \cdots \times_{p+1} \mathbf{U}_p$ |

of $\mathbf{k_x} \in \mathbb{R}^N$ is $\langle \phi(\mathbf{x}^{(n)}), \phi(\mathbf{x}) \rangle = k(\mathbf{x}^{(n)}, \mathbf{x})$ and the tensor $\boldsymbol{\mathcal{C}} \in \mathbb{R}^{N \times d_1 \times \cdots \times d_p}$ is defined by $\boldsymbol{\mathcal{C}} = \boldsymbol{\mathcal{G}} \times_1 \mathbf{A} \times_2 \mathbf{U}_1 \times_2 \cdots \times_{p+1} \mathbf{U}_p$, with $\boldsymbol{\mathcal{G}} = \boldsymbol{\mathcal{Y}} \times_1 \left(\mathbf{A}^\top \mathbf{K}(\mathbf{K} + \gamma \mathbf{I})\mathbf{A}\right)^{-1} \mathbf{A}^\top \mathbf{K} \times_2 \mathbf{U}_2^\top \times_3 \cdots \times_{p+1} \mathbf{U}_p$. Note that $\boldsymbol{\mathcal{C}}$ has multilinear rank $(R_0, \cdots, R_p)$, hence the low mutlilinear rank constraint on $\boldsymbol{\mathcal{W}}$ in the feature space translates into the low rank structure of the coefficient tensor $\boldsymbol{\mathcal{C}}$.

Let $\mathcal{H}$ be the reproducing kernel Hilbert space associated with the kernel $k$. The overall procedure for kernelized HOLRR is summarized in Algorithm 2. This algorithm returns the tensor $\boldsymbol{\mathcal{C}} \in \mathbb{R}^{N \times d_1 \times \cdots \times d_p}$ defining the regression function $f : \mathbf{x} \mapsto \boldsymbol{\mathcal{C}} \bullet_1 \mathbf{k_x} = \sum_{n=1}^N k(\mathbf{x}, \mathbf{x}^{(n)}) \boldsymbol{\mathcal{C}}^{(n)}$, where $\boldsymbol{\mathcal{C}}^{(n)} = \boldsymbol{\mathcal{C}}_{n:\cdots:} \in \mathbb{R}^{d_1 \times \cdots \times d_p}$.

### 3.3 Theoretical Analysis

**Complexity analysis.** HOLRR is a polynomial time algorithm, more precisely it has a time complexity in $\mathcal{O}((d_0)^3 + N((d_0)^2 + d_0 d_1 \cdots d_p) + \max_{i \geq 0} R_i(d_i)^2 + N d_1 \cdots d_p \max_{i \geq 1} d_i)$. In comparison, LRR has a time complexity in $\mathcal{O}((d_0)^3 + N((d_0)^2 + d_0 d_1 \cdots d_p) + (N + R)(d_1 \cdots d_p)^2)$. Since the complexity of HOLRR only have a linear dependence on the product of the output dimensions instead of a quadratic one for LRR, we can conclude that HOLRR will be more efficient than LRR when the output dimensions $d_1, \cdots, d_p$ are large. It is worth mentioning that the method proposed in [22] to solve a convex relaxation of problem 2 is an iterative algorithm that needs to compute SVDs of matrices of size $d_i \times d_1 \cdots d_{i-1} d_{i+1} \cdots d_p$ for each $0 \leq i \leq p$ *at each iteration*, it is thus computationally more expensive than HOLRR. Moreover, since HOLRR only relies on simple linear algebra tools, readily available methods could be used to further improve the speed of the algorithm, e.g. randomized-SVD [8] and random feature approximation of the kernel function [12, 20].

**Approximation guarantees.** It is easy to check that problem (1) is NP-hard since it generalizes the problem of fitting a Tucker decomposition [9]. The following theorem shows that HOLRR is a $(p + 1)$-approximation algorithm for this problem. This result generalizes the approximation guarantees provided by the truncated HOSVD algorithm for the problem of finding the best low multilinear rank approximation of an arbitrary tensor.

**Theorem 2.** *Let $\boldsymbol{\mathcal{W}}^*$ be a solution of problem (1) and let $\boldsymbol{\mathcal{W}}$ be the regression tensor returned by Algorithm 1. If $\mathcal{L} : \mathbb{R}^{d_0 \times \cdots \times d_p} \to \mathbb{R}$ denotes the objective function of (1) w.r.t. $\boldsymbol{\mathcal{W}}$ then $\mathcal{L}(\boldsymbol{\mathcal{W}}) \leq (p+1)\mathcal{L}(\boldsymbol{\mathcal{W}}^*)$.*

**Generalization Bound.** The following theorem gives an upper bound on the excess-risk for the function class $\mathcal{F} = \{\mathbf{x} \mapsto \boldsymbol{\mathcal{W}} \bullet_1 \mathbf{x} : \text{rank}(\boldsymbol{\mathcal{W}}) \leq (R_0, \cdots, R_p)\}$ of tensor-valued regression functions with bounded multilinear rank. Recall that the expected loss of an hypothesis $h \in \mathcal{F}$ w.r.t. the target function $f^*$ is defined by $R(h) = \mathbb{E}_\mathbf{x}[\mathcal{L}(h(\mathbf{x}), f^*(\mathbf{x}))]$ and its empirical loss by $\hat{R}(h) = \frac{1}{N} \sum_{n=1}^N \mathcal{L}(h(\mathbf{x}^{(n)}), f^*(\mathbf{x}^{(n)}))$.

**Theorem 3.** *Let $\mathcal{L} : \mathbb{R}^{d_1 \times \cdots \times d_p} \to \mathbb{R}$ be a loss function satisfying $\mathcal{L}(\boldsymbol{\mathcal{A}}, \boldsymbol{\mathcal{B}}) = \frac{1}{d_1 \cdots d_p} \sum_{i_1, \cdots, i_p} \ell(\boldsymbol{\mathcal{A}}_{i_1, \cdots, i_p}, \boldsymbol{\mathcal{B}}_{i_1, \cdots, i_p})$ for some loss-function $\ell : \mathbb{R} \to \mathbb{R}^+$ bounded by $M$. Then for any $\delta > 0$, with probability at least $1 - \delta$ over the choice of a sample of size $N$, the follow-*

*ing inequality holds for all $h \in \mathcal{F}$: $R(h) \leq \hat{R}(h) + M\sqrt{2D \log\left(\frac{4e(p+2)d_0 d_1 \cdots d_p}{\max_{i \geq 0} d_i}\right) \log(N)/N} +$*

*$M\sqrt{\log\left(\frac{1}{\delta}\right)/(2N)}$, where $D = R_0 R_1 \cdots R_p + \sum_{i=0}^{p} R_i d_i$.*

*Proof.* (Sketch) The complete proof is given in the supplementary material. It relies on bounding the pseudo-dimension of the class of real-valued functions $\tilde{\mathcal{F}} = \left\{(\mathbf{x}, i_1, \cdots, i_p) \mapsto (\boldsymbol{\mathcal{W}} \bullet_1 \mathbf{x})_{i_1, \cdots, i_p} : \text{rank}(\boldsymbol{\mathcal{W}}) = (R_0, \cdots, R_p)\right\}$. We show that the pseudo-dimension of $\tilde{\mathcal{F}}$ is upper bounded by $(R_0 R_1 \cdots R_p + \sum_{i=0}^{p} R_i d_i) \log\left(\frac{4e(p+2)d_0 d_1 \cdots d_p}{\max_{i \geq 0} d_i}\right)$. This is done by leveraging the following result originally due to [26]: the number of sign patterns of $r$ polynomials, each of degree at most $d$, over $q$ variables is at most $(4edr/q)^q$ for all $r > q > 2$ [25, Theorem 2]. The rest of the proof consists in showing that the risk (resp. empirical risk) of hypothesis in $\mathcal{F}$ and $\tilde{\mathcal{F}}$ are closely related and invoking standard error generalization bounds in terms of the pseudo-dimension [17, Theorem 10.6]. $\qquad\square$

Note that generalization bounds based on the pseudo-dimension for multivariate regression without low-rank constraint would involve a term in $\mathcal{O}(\sqrt{d_0 d_1 \cdots d_p})$. In contrast, the bound from the previous theorem only depends on the product of the output dimensions in a term bounded by $\mathcal{O}(\sqrt{\log(d_1 \cdots d_p)})$. In some sense, taking into account the low multilinear rank of the hypothesis allows us to significantly reduce the dependence on the output dimensions from $\mathcal{O}(\sqrt{d_0 \cdots d_p})$ to $\mathcal{O}(\sqrt{(R_0 \cdots R_p + \sum_i R_i d_i)(\sum_i \log(d_i))})$.

## 4 Experiments

In this section, we evaluate HOLRR on both synthetic and real-world datasets. Our experimental results are for tensor-structured output regression problems on which we report root mean-squared errors (RMSE) averaged across all the outputs. We compare HOLLR with the following methods: regularized least squares **RLS**, low-rank regression **LRR** described in Section 2.2, a multilinear approach based on tensor trace norm regularization **ADMM** [7, 22], a nonconvex multilinear multitask learning approach **MLMT-NC** [22], an higher order extension of partial least squares **HOPLS** [28] and the greedy tensor approach for multivariate spatio-temporal analysis **Greedy** [2].

For experiments with kernel algorithms we use the readily available kernelized RLS and the LRR kernel extension proposed in [18]. Note that ADMM, MLMT-NC and Greedy only consider a linear dependency between inputs and outputs. The greedy tensor algorithm proposed in [2] is developed specially for spatio-temporal data and the implementation provided by the authors is restricted to third-order tensors. Although MLMLT-NC is perhaps the closest algorithm to ours, we applied it only to simulated data. This is because MLMLT-NC is computationally very expensive and becomes intractable for large data sets. Average running times are reported in supplementary material B.

### 4.1 Synthetic Data

We generate both linear and nonlinear data. Linear data is drawn from the model $\boldsymbol{\mathcal{Y}} = \boldsymbol{\mathcal{W}} \bullet_1 \mathbf{x} + \boldsymbol{\mathcal{E}}$ where $\boldsymbol{\mathcal{W}} \in \mathbb{R}^{10 \times 10 \times 10 \times 10}$ is a tensor of multilinear rank $(6, 4, 4, 8)$ drawn at random, $\mathbf{x} \in \mathbb{R}^{10}$ is drawn from $\mathcal{N}(0, \mathbf{I})$, and each component of the error tensor $\boldsymbol{\mathcal{E}}$ is drawn from $\mathcal{N}(0, 0.1)$. Nonlinear data is drawn from $\boldsymbol{\mathcal{Y}} = \boldsymbol{\mathcal{W}} \bullet_1 (\mathbf{x} \otimes \mathbf{x}) + \boldsymbol{\mathcal{E}}$ where $\boldsymbol{\mathcal{W}} \in \mathbb{R}^{25 \times 10 \times 10 \times 10}$ is of rank $(5, 6, 4, 2)$ and $\mathbf{x} \in \mathbb{R}^5$ and $\boldsymbol{\mathcal{E}}$ are generated as above. Hyper-parameters for all algorithms are selected using 3-fold cross-validation on the training data.

These experiments have been carried out for different sizes of the training data set, 20 trials have been executed for each size. The average RMSEs on a test set of size 100 for the 20 trials are reported in Figure 2. We see that HOLRR algorithm clearly outperforms the other methods on the linear data. MLMT-NC method achieved the second best performance, it is however much more computationally expensive (see Table 1 in supplementary material B). On the nonlinear data LRR achieves good performances but HOLRR is still significantly more accurate, especially with small training datasets.

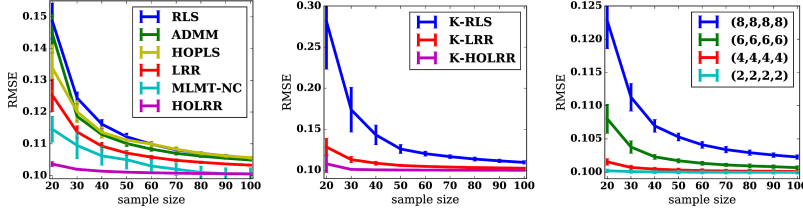

Figure 2: Average RMSE as a function of the training set size: (left) linear data, (middle) nonlinear data, (right) for different values of the rank parameter.

Table 1: RMSE on forecasting task.

| Data set | ADMM | Greedy | HOPLS | HOLRR | K-HOLRR (poly) | K-HOLRR (rbf) |
|---|---|---|---|---|---|---|
| CCDS | 0.8448 | 0.8325 | 0.8147 | 0.8096 | 0.8275 | **0.7913** |
| Foursquare | 0.1407 | **0.1223** | **0.1224** | 0.1227 | **0.1223** | 0.1226 |
| Meteo-UK | 0.6140 | – | 0.625 | 0.5971 | 0.6107 | **0.5886** |

To see how sensitive HOLLR is w.r.t. the choice of the multilinear rank, we carried out a similar experiment comparing HOLLR performances for different values of the rank parameter, see Fig. 2 (right). In this experiment, the rank of the tensor $\mathcal{W}$ used to generate the data is $(2, 2, 2, 2)$ while the input and output dimensions and the noise level are the same as above.

## 4.2 Real Data

We evaluate our algorithm on a forecasting task on the following real-world data sets:
**CCDS:** the comprehensive climate data set is a collection of climate records of North America from [15]. The data set contains monthly observations of 17 variables such as Carbon dioxide and temperature spanning from 1990 to 2001 across 125 observation locations.
**Foursquare:** the Foursquare data set [14] contains users' check-in records in Pittsburgh area categorized by different venue types such as Art & University. It records the number of check-ins by 121 users in each of the 15 category of venues over 1200 time intervals.
**Meteo-UK:** The data set is collected from the meteorological office of the UK[2]. It contains monthly measurements of 5 variables in 16 stations across the UK from 1960 to 2000.

The forecasting task consists in predicting all variables at times $t + 1, \ldots, t + k$ from their values at times $t - 2$, $t - 1$ and $t$. The first two real data sets were used in [2] with $k = 1$ (i.e. outputs are matrices). We consider here the same setting for these two data sets. For the third dataset we consider higher-order output tensors by setting $k = 5$. The output tensors are thus of size respectively $17 \times 125$, $15 \times 121$ and $16 \times 5 \times 5$ for the three datasets.

For all the experiments, we use 90% of the available data for training and 10% for testing. All hyper-parameters are chosen by cross-validation. The average test RMSE over 10 runs are reported in Table 1 (running times are reported in Table 1 in supplementary material B). We see that HOLRR and K-HOLRR outperforms the other methods on the CCDS dataset while being orders of magnitude faster for the kernelized version (0.61s vs. 75.47s for Greedy and 235.73s for ADMM in average). On the Foursquare dataset HOLRR performs as well as Greedy and on the Meteo-UK dataset K-HOLRR gets the best results with the RBF kernel while being much faster than ADMM (1.66s vs. 40.23s in average).

## 5 Conclusion

We proposed a low-rank multilinear regression model for tensor-structured output data. We developed a fast and efficient algorithm to tackle the multilinear rank penalized minimization problem and provided theoretical guarantees. Experimental results showed that capturing low-rank structure in the output data can help to improve tensor regression performance.

**Acknowledgments**

We thank François Denis and the reviewers for their helpful comments and suggestions. This work was partially supported by ANR JCJC program MAD (ANR- 14-CE27-0002).

## Footnotes

[1] An extended version of this experiment is presented in supplementary material B.

[2] http://www.metoffice.gov.uk/public/weather/climate-historic/

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
