[Supplementary Material · supplementary.pdf]

# Low-Rank Regression with Tensor Responses (Supplementary Material)

**Guillaume Rabusseau and Hachem Kadri**
Aix Marseille Univ, CNRS, LIF, Marseille, France
{firstname.lastname}@lif.univ-mrs.fr

## A  Proofs

### A.1  Proof of Theorem 1

**Theorem.** *For given orthogonal matrices* $\mathbf{U}_0, \cdots, \mathbf{U}_p$ *the tensor* $\boldsymbol{\mathcal{G}}$ *that minimizes (2) is given by*

$$\boldsymbol{\mathcal{G}} = \tilde{\boldsymbol{\mathcal{Y}}} \times_1 (\mathbf{U}_0^\top \tilde{\mathbf{X}}^\top \tilde{\mathbf{X}} \mathbf{U}_0)^{-1} \mathbf{U}_0^\top \tilde{\mathbf{X}}^\top \times_2 \mathbf{U}_1^\top \times_3 \cdots \times_{p+1} \mathbf{U}_p^\top \ .$$

*Proof.* Since the Frobenius norm of a tensor is equal to the one of its vectorization the objective function in (2) can be written as

$$\|(\mathbf{U}_p \otimes \mathbf{U}_{p-1} \otimes \cdots \otimes \mathbf{U}_1 \otimes \tilde{\mathbf{X}} \mathbf{U}_0)\mathrm{vec}(\boldsymbol{\mathcal{G}}) - \mathrm{vec}(\tilde{\boldsymbol{\mathcal{Y}}})\|_F^2 \ .$$

Let $\mathbf{M} = \mathbf{U}_p \otimes \mathbf{U}_{p-1} \otimes \cdots \otimes \mathbf{U}_1 \otimes \tilde{\mathbf{X}} \mathbf{U}_0$. The solution w.r.t. $\mathrm{vec}(\boldsymbol{\mathcal{G}})$ of this classical linear least-squares problem is given by $(\mathbf{M}^\top \mathbf{M})^{-1} \mathbf{M}^\top$. Using the mixed-product and inverse properties of the Kronecker product and the column-wise orthogonality of $\mathbf{U}_1, \cdots, \mathbf{U}_p$ we obtain $\mathrm{vec}(\boldsymbol{\mathcal{G}}) = \left(\mathbf{U}_p \otimes \cdots \otimes \mathbf{U}_1 \otimes (\mathbf{U}_0^\top \tilde{\mathbf{X}}^\top \tilde{\mathbf{X}} \mathbf{U}_0)^{-1} \mathbf{U}_0^\top \tilde{\mathbf{X}}^\top\right) \mathrm{vec}(\tilde{\boldsymbol{\mathcal{Y}}})$.  □

### A.2  Proof of Proposition 1

**Proposition.** *For* $0 \le i \le p$, *using the definition of* $\boldsymbol{\Pi}_i$ *in (3), the optimal solution of*

$$\min_{\mathbf{U}_i \in \mathbb{R}^{d_i \times R_i}} \|\tilde{\boldsymbol{\mathcal{Y}}} \times_{i+1} \boldsymbol{\Pi}_i - \tilde{\boldsymbol{\mathcal{Y}}}\|_F^2 \ s.t. \ \mathbf{U}_i^\top \mathbf{U}_i = \mathbf{I}$$

*is given by the eigenvectors of*

$$\begin{cases} (\tilde{\mathbf{X}}^\top \tilde{\mathbf{X}})^{-1} \tilde{\mathbf{X}}^\top \tilde{\mathbf{Y}}_{(1)} \tilde{\mathbf{Y}}_{(1)}^\top \tilde{\mathbf{X}} & \textit{if } i = 0 \\ \tilde{\mathbf{Y}}_{(i)} \tilde{\mathbf{Y}}_{(i)}^\top & \textit{otherwise} \end{cases}$$

*that corresponds to the* $R_i$ *largest eigenvalues.*

*Proof.* For any $0 \le i \le p$, since $\boldsymbol{\Pi}_i$ is a projection we have $\langle \tilde{\boldsymbol{\mathcal{Y}}} \times_1 \boldsymbol{\Pi}_i, \tilde{\boldsymbol{\mathcal{Y}}} \rangle = \langle \boldsymbol{\Pi}_i \tilde{\mathbf{Y}}_{(i)}, \tilde{\mathbf{Y}}_{(i)} \rangle = \|\boldsymbol{\Pi}_i \tilde{\mathbf{Y}}_{(i)}\|_F^2$, thus minimizing $\|\tilde{\boldsymbol{\mathcal{Y}}} \times_i \boldsymbol{\Pi}_i - \tilde{\boldsymbol{\mathcal{Y}}}\|_F^2$ is equivalent to minimizing $\|\boldsymbol{\Pi}_i \tilde{\mathbf{Y}}_{(i)}\|_F^2 - 2\langle \boldsymbol{\Pi}_i \tilde{\mathbf{Y}}_{(i)}, \tilde{\mathbf{Y}}_{(i)} \rangle = -\|\boldsymbol{\Pi}_i \tilde{\mathbf{Y}}_{(i)}\|_F^2$. For $i \ge 1$, we have $\|\boldsymbol{\Pi}_i \tilde{\mathbf{Y}}_{(i)}\|_F^2 = \mathrm{Tr}(\mathbf{U}_i^\top \tilde{\mathbf{Y}}_{(i)} \tilde{\mathbf{Y}}_{(i)}^\top \mathbf{U}_i)$ which is maximized by letting the columns of $\mathbf{U}_i$ be the top $R_i$ eigenvectors of the matrix $\tilde{\mathbf{Y}}_{(i)} \tilde{\mathbf{Y}}_{(i)}^\top$. For $i = 0$ we have $\|\boldsymbol{\Pi}_0 \tilde{\mathbf{Y}}_{(i)}\|_F^2 = \mathrm{Tr}(\boldsymbol{\Pi}_0 \tilde{\mathbf{Y}}_{(1)} \tilde{\mathbf{Y}}_{(1)}^\top \boldsymbol{\Pi}_0^\top) = \mathrm{Tr}\left((\mathbf{U}_0^\top \mathbf{A} \mathbf{U}_0)^{-1} \mathbf{U}_0^\top \mathbf{B} \mathbf{U}_0\right)$ with $\mathbf{A} = \tilde{\mathbf{X}}^\top \tilde{\mathbf{X}}$ and $\mathbf{B} = \tilde{\mathbf{X}}^\top \tilde{\mathbf{Y}}_{(1)} \tilde{\mathbf{Y}}_{(1)}^\top \tilde{\mathbf{X}}$, which is maximized by the top $R_0$ eigenvectors of $\mathbf{A}^{-1}\mathbf{B}$.  □

## A.3 Proof of Proposition 2

**Proposition.** *If $\boldsymbol{\alpha} \in \mathbb{R}^N$ is an eigenvector with eigenvalue $\lambda$ of the matrix*

$$(\mathbf{K} + \gamma \mathbf{I})^{-1} \mathbf{Y}_{(1)} \mathbf{Y}_{(1)}^\top \mathbf{K} \ ,$$

*then $\mathbf{v} = \boldsymbol{\Phi}^\top \boldsymbol{\alpha} \in \mathbb{R}^L$ is an eigenvector with eigenvalue $\lambda$ of the matrix $(\boldsymbol{\Phi}^\top \boldsymbol{\Phi} + \gamma \mathbf{I})^{-1} \boldsymbol{\Phi}^\top \mathbf{Y}_{(1)} \mathbf{Y}_{(1)}^\top \boldsymbol{\Phi}$.*

*Proof.* Let $\boldsymbol{\alpha} \in \mathbb{R}^N$ be the eigenvector from the hypothesis. We have

$$
\begin{aligned}
\lambda \mathbf{v} = \boldsymbol{\Phi}^\top (\lambda \boldsymbol{\alpha}) &= \boldsymbol{\Phi}^\top \left( (\mathbf{K} + \gamma \mathbf{I})^{-1} \mathbf{Y}_{(1)} \mathbf{Y}_{(1)}^\top \mathbf{K} \right) \boldsymbol{\alpha} \\
&= \boldsymbol{\Phi}^\top (\boldsymbol{\Phi} \boldsymbol{\Phi}^\top + \gamma \mathbf{I})^{-1} \mathbf{Y}_{(1)} \mathbf{Y}_{(1)}^\top \boldsymbol{\Phi} \boldsymbol{\Phi}^\top \boldsymbol{\alpha} \\
&= \left( (\boldsymbol{\Phi}^\top \boldsymbol{\Phi} + \gamma \mathbf{I})^{-1} \boldsymbol{\Phi}^\top \mathbf{Y}_{(1)} \mathbf{Y}_{(1)}^\top \boldsymbol{\Phi} \right) \mathbf{v} \ . \qquad \square
\end{aligned}
$$

## A.4 Proof of Theorem 2

**Theorem.** *Let $\boldsymbol{\mathcal{W}}^*$ be a solution of problem (1) and let $\boldsymbol{\mathcal{W}}$ be the regression tensor returned by Algorithm 1. If $\mathcal{L} : \mathbb{R}^{d_0 \times \cdots \times d_p} \to \mathbb{R}$ denotes the objective function of (1) with respect to $\boldsymbol{\mathcal{W}}$ then*

$$\mathcal{L}(\boldsymbol{\mathcal{W}}) \leq (p+1) \mathcal{L}(\boldsymbol{\mathcal{W}}^*).$$

The proof of this theorem relies on the following lemma which was proved in [1] to obtain a nice and elegant proof for the approximation guarantees of the HOSVD algorithm for the problem of low multilinear rank approximation of a given tensor.

**Lemma 1.** *Let $\boldsymbol{\mathcal{T}} \in \mathbb{R}^{d_1 \times \cdots \times d_p}$ be a pth order tensor, let $m, n \in [p]$, and let $\mathbf{P} \in \mathbb{R}^{d_m \times d_m}$ and $\mathbf{Q} \in \mathbb{R}^{d_n \times d_n}$ be two orthogonal projection matrices. Then*

$$\| \boldsymbol{\mathcal{T}} - \boldsymbol{\mathcal{T}} \times_m \mathbf{P} \times_n \mathbf{Q} \|_F^2 \leq \| \boldsymbol{\mathcal{T}} - \boldsymbol{\mathcal{T}} \times_m \mathbf{P} \|_F^2 + \| \boldsymbol{\mathcal{T}} - \boldsymbol{\mathcal{T}} \times_n \mathbf{Q} \|_F^2.$$

*Proof.* First observe that for any orthogonal projection matrix $\boldsymbol{\Pi}$ and any tensors $\boldsymbol{\mathcal{A}}, \boldsymbol{\mathcal{B}}$ we have

$$\| \boldsymbol{\mathcal{A}} \times_n \boldsymbol{\Pi} \|_F^2 \leq \| \boldsymbol{\mathcal{A}} \|_F^2 \quad \text{and} \quad \| \boldsymbol{\mathcal{A}} \times_n (\mathbf{I} - \boldsymbol{\Pi}) + \boldsymbol{\mathcal{B}} \times_n \boldsymbol{\Pi} \|_F^2 = \| \boldsymbol{\mathcal{A}} \times_n (\mathbf{I} - \boldsymbol{\Pi}) \|_F^2 + \| \boldsymbol{\mathcal{B}} \times_n \boldsymbol{\Pi} \|_F^2.$$

Both equations follow from the fact that the Frobenius norm of a tensor is equal to the one of any of its matricization. Indeed

$$\| \boldsymbol{\mathcal{A}} \times_n \boldsymbol{\Pi} \|_F^2 = \| \boldsymbol{\Pi} \mathbf{A}_{(n)} \|_F^2 \leq \| \mathbf{A}_{(n)} \|_F^2 = \| \boldsymbol{\mathcal{A}} \|_F^2$$

since $\boldsymbol{\Pi}$ is a projection. The second equality is proved similarly using the orthogonality of $\boldsymbol{\Pi}$ and $\mathbf{I} - \boldsymbol{\Pi}$.

Then, under the hypothesis of the lemma, we have

$$
\begin{aligned}
\| \boldsymbol{\mathcal{T}} - \boldsymbol{\mathcal{T}} \times_m \mathbf{P} \times_n \mathbf{Q} \|_F^2 &= \| \boldsymbol{\mathcal{T}} \times_m (\mathbf{I} - \mathbf{P}) + (\boldsymbol{\mathcal{T}} - \boldsymbol{\mathcal{T}} \times_n \mathbf{Q}) \times_m \mathbf{P} \|_F^2 \\
&= \| \boldsymbol{\mathcal{T}} \times_m (\mathbf{I} - \mathbf{P}) \|_F^2 + \| (\boldsymbol{\mathcal{T}} - \boldsymbol{\mathcal{T}} \times_n \mathbf{Q}) \times_m \mathbf{P} \|_F^2 \\
&\leq \| \boldsymbol{\mathcal{T}} - \boldsymbol{\mathcal{T}} \times_m \mathbf{P} \|_F^2 + \| \boldsymbol{\mathcal{T}} - \boldsymbol{\mathcal{T}} \times_n \mathbf{Q} \|_F^2. \qquad \square
\end{aligned}
$$

Let $\mathbf{U}_0, \cdots, \mathbf{U}_p$ be the matrices defined in Algorithm 1 and let $\boldsymbol{\Pi}_0, \cdots, \boldsymbol{\Pi}_p$ be the orthogonal projection matrices defined in problem (3). The regression tensor $\boldsymbol{\mathcal{W}}$ returned by HOLRR satisfies

$$\boldsymbol{\mathcal{W}} \times_1 \tilde{\mathbf{X}} = \tilde{\boldsymbol{\mathcal{Y}}} \times_1 \boldsymbol{\Pi}_0 \times_2 \cdots \times_{p+1} \boldsymbol{\Pi}_p.$$

Similarly, it follows from Theorem 1 that a solution $\boldsymbol{\mathcal{W}}^*$ of problem (1) satisfies

$$\boldsymbol{\mathcal{W}}^* \times_1 \tilde{\mathbf{X}} = \tilde{\boldsymbol{\mathcal{Y}}} \times_1 \boldsymbol{\Pi}_0^* \times_2 \cdots \times_{p+1} \boldsymbol{\Pi}_p^*$$

for some orthogonal projection matrices $\boldsymbol{\Pi}_i^*$ for $0 \leq i \leq p$.

Using successive applications of the previous Lemma we obtain

$$\mathcal{L}(\boldsymbol{\mathcal{W}}) = \|\boldsymbol{\mathcal{W}} \times_1 \tilde{\mathbf{X}} - \tilde{\boldsymbol{\mathcal{Y}}}\|_F^2 = \|\tilde{\boldsymbol{\mathcal{Y}}} \times_1 \boldsymbol{\Pi}_0 \times_2 \cdots \times_{p+1} \boldsymbol{\Pi}_p - \tilde{\boldsymbol{\mathcal{Y}}}\|_F^2 \leq \sum_{i=0}^p \|\tilde{\boldsymbol{\mathcal{Y}}} \times_{i+1} \boldsymbol{\Pi}_i - \tilde{\boldsymbol{\mathcal{Y}}}\|_F^2.$$

By Proposition 1, each summand in this upper bound is minimal with respect to $\boldsymbol{\Pi}_i$, hence $\|\tilde{\boldsymbol{\mathcal{Y}}} \times_{i+1} \boldsymbol{\Pi}_i - \tilde{\boldsymbol{\mathcal{Y}}}\|_F^2 \leq \|\tilde{\boldsymbol{\mathcal{Y}}} \times_{i+1} \boldsymbol{\Pi}_i^* - \tilde{\boldsymbol{\mathcal{Y}}}\|_F^2$ for any $i \in [p]$. It remains to show that

$$\|\tilde{\boldsymbol{\mathcal{Y}}} \times_{i+1} \boldsymbol{\Pi}_i^* - \tilde{\boldsymbol{\mathcal{Y}}}\|_F^2 \leq \|\tilde{\boldsymbol{\mathcal{Y}}} \times_1 \boldsymbol{\Pi}_0^* \times_2 \cdots \times_{p+1} \boldsymbol{\Pi}_p^* - \tilde{\boldsymbol{\mathcal{Y}}}\|_F^2 = \mathcal{L}(\boldsymbol{\mathcal{W}}^*)$$

for all $i \in [p]$. Indeed, using the fact that the Frobenius norm of a tensor is equal to the one of its matricization, we obtain for the case $i = 0$

$$\begin{aligned}
\|\tilde{\boldsymbol{\mathcal{Y}}} \times_1 \boldsymbol{\Pi}_0^* \times_2 \cdots \times_{p+1} \boldsymbol{\Pi}_p^* - \tilde{\boldsymbol{\mathcal{Y}}}\|^2 &= \|\boldsymbol{\Pi}_0^* \tilde{\mathbf{Y}}_{(1)} (\boldsymbol{\Pi}_p^* \otimes \cdots \otimes \boldsymbol{\Pi}_1^*)^\top - \tilde{\mathbf{Y}}_{(1)}\|_F^2 \\
&= \|(\boldsymbol{\Pi}_0^* - \mathbf{I}_{d_0}) \tilde{\mathbf{Y}}_{(1)} + \boldsymbol{\Pi}_0^* \tilde{\mathbf{Y}}_{(1)} (\boldsymbol{\Pi}_p^* \otimes \cdots \otimes \boldsymbol{\Pi}_1^* - \mathbf{I}_{d_1 d_2 \cdots d_p})^\top\|_F^2 \\
&= \|(\boldsymbol{\Pi}_0^* - \mathbf{I}_{d_0}) \tilde{\mathbf{Y}}_{(1)}\|_F^2 + \|\boldsymbol{\Pi}_0^* \tilde{\mathbf{Y}}_{(1)} (\boldsymbol{\Pi}_p^* \otimes \cdots \otimes \boldsymbol{\Pi}_1^* - \mathbf{I}_{d_1 d_2 \cdots d_p})^\top\|_F^2 \\
&\geq \|(\boldsymbol{\Pi}_0^* - \mathbf{I}_{d_0}) \tilde{\mathbf{Y}}_{(1)}\|_F^2 \\
&= \|\tilde{\boldsymbol{\mathcal{Y}}} \times_1 \boldsymbol{\Pi}_0^* - \tilde{\boldsymbol{\mathcal{Y}}}\|_F^2
\end{aligned}$$

where we used the orthogonality of $\boldsymbol{\Pi}_0^*$ and $\boldsymbol{\Pi}_0^* - \mathbf{I}_{d_0}$. The proofs for other values of $i$ are similar.

### A.5    Proof of Theorem 3

We start by bounding the pseudo-dimension of the class of real-valued functions with domain $\mathbb{R}^{d_0} \times [d_1] \times \cdots \times [d_p]$

$$\tilde{\mathcal{F}} = \left\{ (\mathbf{x}, i_1, \cdots, i_p) \mapsto (\boldsymbol{\mathcal{W}} \bullet_1 \mathbf{x})_{i_1, \cdots, i_p} \; : \; \mathrm{rank}(\boldsymbol{\mathcal{W}}) = (R_0, \cdots, R_p) \right\}.$$

We first recall the definition of the pseudo-dimension of a class of real-valued functions.

**Definition 1.** *A class $\mathcal{F}$ of real-valued functions pseudo-shatters the points $x_1, \cdots, x_m$ with thresholds $t_1, \cdots, t_m$ if for every binary labeling of the points $(s_1, \cdots, s_m) \in \{-, +\}^m$ there exists $f \in \mathcal{F}$ s.t. $f(x_i) < t_i$ iff $s_i = -$. The pseudo-dimension of a class $F$ is the supremum over $m$ for which there exist $m$ points that are pseudo-shattered by $\mathcal{F}$ (with some thresholds).*

We say that a set of polynomials $p_1, p_2, \cdots, p_k$ has at least $m$ sign patterns if there exist $x_1, \cdots, x_m$ such that such that the sign vectors $\mathbf{v}_i = [sign(p_1(x_i)), \cdots, sign(p_k(x_i))]^\top$ are pairwise distinct. Following [4], the following theorem bounds the number of sign patterns for a set of polynomials.

**Theorem.** *[3, Theorem 34, 35] The number of sign patterns of $r$ polynomials, each of degree at most $d$, over $q$ variables is at most $\left( \frac{4edr}{q} \right)^q$ for all $r > q > 2$.*

The following lemma gives an upper bound on the pseudo-dimension of $\tilde{\mathcal{F}}$ using the previous theorem.

**Lemma 2.** *The pseudo-dimension of the real-valued function class $\tilde{\mathcal{F}}$ is upper bounded by $(R_0 R_1 \cdots R_p + \sum_{i=0}^p R_i d_i) \log \left( \frac{4e(p+2)d_0 d_1 \cdots d_p}{d_0 + d_1 + \cdots + d_p} \right)$.*

*Proof.* It is well known that the pseudo-dimension of a vector space of real-valued functions is equal to its dimension [2, Theorem 10.5]. Since $\tilde{\mathcal{F}}$ is a (non-linear) subspace of the $d_0 d_1 \cdots d_p$-dimensional vector space

$$\left\{ (\mathbf{x}, i_1, \cdots, i_p) \mapsto (\boldsymbol{\mathcal{W}} \bullet_1 \mathbf{x})_{i_1, \cdots, i_p} \; : \; \boldsymbol{\mathcal{W}} \in \mathbb{R}^{d_0 \times \cdots \times d_p} \right\}$$

of real-valued functions with domain $\mathbb{R}^{d_0} \times [d_1] \times \cdots \times [d_p]$, the pseudo-dimension of $\tilde{\mathcal{F}}$ is bounded by $d_0 d_1 \cdots d_p$.

Now, let $m \leq d_0 \cdots d_p$ and let $\{(\mathbf{x}^k, i_1^k, \cdots, i_p^k)\}_{k=1}^m$ be a set of points that are pseudo-shattered by $\tilde{\mathcal{F}}$ with thresholds $t_1, \cdots, t_m \in \mathbb{R}$. Then for each sign pattern $(s_1, \cdots, s_m) \in$

$\{-, +\}^m$, there exists $\tilde{f} \in \tilde{\mathcal{F}}$ such that $sign(\tilde{f}(\mathbf{x}^k, i_1^k, \cdots, i_p^k) - t_k) = s_k$. Any function $\tilde{f} \in \tilde{\mathcal{F}}$ can be written as

$$(\mathbf{x}, j_1, \cdots, j_p) \mapsto \left(\boldsymbol{\mathcal{G}} \times_1 \mathbf{x}^\top \mathbf{U}_0 \times_2 \mathbf{U}_1 \cdots \times_{p+1} \mathbf{U}_p\right)_{j_1, \cdots, j_p}$$

for some $\boldsymbol{\mathcal{G}} \in \mathbb{R}^{R_0 \times \cdots \times R_p}$, $\mathbf{U}_i \in \mathbb{R}^{d_i \times R_i}$ for $0 \le i \le p$. Thus, considering the entries of $\boldsymbol{\mathcal{G}}, \mathbf{U}_0, \cdots, \mathbf{U}_p$ as variables, the set $\{\tilde{f}(\mathbf{x}^k, i_1^k, \cdots, i_p^k) - t_k\}_{k=1}^m$ can be seen as a set of $m$ polynomials of degree at most $p + 2$ over these $D = R_0 \cdots R_p + \sum_{i=0}^{p} d_i R_i$ variables. It then follows from the previous theorem that $2^m \le \left(\frac{4e(p+2)m}{D}\right)^D$. The result follows using $m \le d_0 \cdots d_p$ and $D \ge \sum_{i=0}^{p} d_i$.

$\square$

Once the pseudo-dimension of the function class $\tilde{\mathcal{F}}$ is bounded, one can invoke standard error generalization bounds in terms of the pseudo-dimension [2, Theorem 10.6] to obtain the following theorem that gives an upper bound on the excess risk for the class of function

$$\mathcal{F} = \{\mathbf{x} \mapsto \boldsymbol{\mathcal{W}} \bullet_1 \mathbf{x} \ : \ \text{rank}(\boldsymbol{\mathcal{W}}) = (R_0, \cdots, R_p)\}.$$

**Theorem.** *Let $\mathcal{L} : \mathbb{R}^{d_1 \times \cdots \times d_p} \to \mathbb{R}$ be a loss function satisfying*

$$\mathcal{L}(\boldsymbol{\mathcal{A}}, \boldsymbol{\mathcal{B}}) = \frac{1}{d_1 \cdots d_p} \sum_{i_1, \cdots, i_p} \ell(\boldsymbol{\mathcal{A}}_{i_1, \cdots, i_p}, \boldsymbol{\mathcal{B}}_{i_1, \cdots, i_p})$$

*for some loss function $\ell : \mathbb{R} \to \mathbb{R}^+$ bounded by $M$. Then for any $\delta > 0$, with probability at least $1 - \delta$ over the choice of a sample of size $N$, the following inequality holds for all $h \in \mathcal{F}$:*

$$R(h) \le \hat{R}(h) + M\sqrt{\frac{2D \log\left(\frac{4e(p+2)d_0 d_1 \cdots d_p}{d_0 + d_1 + \cdots + d_p}\right) \log N}{N}} + M\sqrt{\frac{\log\left(\frac{1}{\delta}\right)}{2N}}$$

*where $D = R_0 R_1 \cdots R_p + \sum_{i=0}^{p} R_i d_i$.*

*Proof.* For any $h : \mathbb{R}^{d_0} \to \mathbb{R}^{d_1 \times \cdots \times d_p}$ we define $\tilde{h} : \mathbb{R}^{d_0} \times [d_1] \times \cdots \times [d_p] \to \mathbb{R}$ by $\tilde{h}(\mathbf{x}, i_1, \cdots, i_p) = h(\mathbf{x})_{i_1 \cdots i_p}$. Let $\mathcal{D}$ denote the distribution of the input data. We have

$$R(h) = \mathop{\mathbb{E}}_{\mathbf{x} \sim \mathcal{D}}[\mathcal{L}(f(\mathbf{x}), h(\mathbf{x}))] = \frac{1}{d_1 \cdots d_p} \sum_{i_1, \cdots, i_p} \mathop{\mathbb{E}}_{\mathbf{x} \sim \mathcal{D}}[\ell(f(\mathbf{x})_{i_1 \cdots i_p}, h(\mathbf{x})_{i_1 \cdots i_p})]$$

$$= \mathop{\mathbb{E}}_{\substack{\mathbf{x} \sim \mathcal{D} \\ i_k \sim \mathcal{U}(d_k), k \in [p]}} [\ell(\tilde{f}(\mathbf{x}, i_1, \cdots, i_p), \tilde{h}(\mathbf{x}, i_1, \cdots, i_p))]$$

where $\mathcal{U}(k)$ denotes the discrete uniform distribution on $[k]$ for any integer $k \ge 1$. It follows that $R(h) = R(\tilde{h})$. Similarly, one can show that $\hat{R}(h) = \hat{R}(\tilde{h})$. The result then directly follows using Theorem 10.6 in [2] (see below) to bound $R(\tilde{h}) - \hat{R}(\tilde{h})$. $\square$

**Theorem** (Theorem 10.6 in [2])**.** *Let $H$ be a family of real-valued functions and let $G = \{x \mapsto L(h(x), f(x)) : h \in H\}$ be the family of loss functions associated to $H$. Assume that the pseudo-dimension of $G$ is bounded by $d$ and that the loss function $L$ is bounded by $M$. Then, for any $\delta > 0$, with probability at least $\delta$ over the choice of a sample of size $m$, the following inequality holds for all $h \in H$:*

$$R(h) \le \hat{R}(h) + M\sqrt{\frac{2d \log\left(\frac{em}{d}\right)}{m}} + M\sqrt{\frac{\log\left(\frac{1}{\delta}\right)}{2m}}.$$

# B  Experiments

## B.1  Running Times

The running times of different tensor response regression algorithms on synthetic and real data sets are given in Table 1.

Table 1: Average running times in seconds for some of the experiments. We did not run MLMT-NC on the real world data sets because it is computationally very expensive. The implementation of the Greedy algorithm is limited to 2nd order output tensors, this is why we did not run it on the synthetic and Meteo UK data sets. Finally, the synthetic non linear data was generated using a polynomial relation which is why the RBF kernel was not used on this data set.

| Data set | MLMTL-NC | ADMM | Greedy | HOPLS | HOLRR | K-HOLRR (poly) | K-HOLRR (rbf) |
|---|---|---|---|---|---|---|---|
| Synthetic | 945.79 | 12.92 | – | 0.12 | 0.04 | 0.53 | – |
| CCDS | – | 235.73 | 75.47 | 121.28 | 100.94 | 0.46 | 0.61 |
| Foursquare | – | 33.83 | 37.70 | 22.3 | 14.41 | 19.20 | 19.67 |
| Meteo UK | – | 40.23 | – | 2.12 | 1.67 | 1.57 | 1.66 |

## B.2 Image Reconstruction from Noisy Measurements

To give an illustrative intuition on the differences between matrix and multilinear rank regularization we generate data from the model $\mathcal{Y} = \mathcal{W} \bullet_1 \mathbf{x} + \mathcal{E}$ where the tensor $\mathcal{W}$ is a color image of size $m \times n$ encoded with three color channels RGB. We consider two different tasks depending on the input dimension: (i) $\mathcal{W} \in \mathbb{R}^{3 \times m \times n}$, $\mathbf{x} \in \mathbb{R}^3$ and (ii) $\mathcal{W} \in \mathbb{R}^{n \times m \times 3}$, $\mathbf{x} \in \mathbb{R}^n$. In both tasks the components of both $\mathbf{x}$ and $\mathcal{E}$ are drawn from $\mathcal{N}(0,1)$ and the regression tensor $\mathcal{W}$ is learned from a training set of size 200.

This experiment allows us to visualize the tensors returned by the RLS, LRR and HOLRR algorithms. The results are shown in Figure 1 for three images: a green cross (of size $50 \times 50$), a thumbnail of a Rothko painting ($44 \times 70$) and a square made of triangles ($70 \times 70$), note that the first two images have a low rank structure which is not the case for the third one.

We first see that HOLRR clearly outperforms LRR on the task where the input dimension is small (task (i)). This is to be expected since the rank of the matrix $\mathbf{W}_{(1)}$ is at most 3 and LRR is unable to enforce a low-rank structure on the output modes of $\mathcal{W}$. When the rank constraint is set to 1 for LRR and $(3,1,1)$ for HOLRR, we clearly see that (unlike HOLRR) the LRR approach does not enforce any low-rank structure on the regression tensor along the output modes. On task (ii) the difference is more subtle, but we can see that setting a rank constraint of 2 for the LRR algorithm prevents the model from capturing the white border around the green cross and creates the vertical lines artifact in the Rothko painting. For higher values of the rank the model starts to learn the noise. The tensor returned by HOLRR with rank $(2,2,3)$ for the cross image and $(4,4,3)$ for the Rothko painting do not exhibit these behaviors and give better results on these two images. On the square image which does not have a low-rank structure both algorithms exhibit underfitting for low values of the rank parameter. Overall, we see that capturing the multilinear low-rank structure of the output data allows HOLRR to separate the noise from the true signal better than RLS and LRR.

Figure 1: Image reconstruction from noisy measurements: $\boldsymbol{\mathcal{Y}} = \boldsymbol{\mathcal{W}} \bullet_1 \mathbf{x} + \boldsymbol{\mathcal{E}}$ where $\boldsymbol{\mathcal{W}}$ is a color image (RGB). (left) Task (i): input dimension is the number of channels. (right) Task (ii): input dimension is the height of the image. Each image is labeled with the name of the algorithm followed by the value used for the rank constraint.