[Reviews · NeurIPS 2016]

Reviewer 1

Summary

The paper presents an efficient spectral algorithm for approximating the solution of regression problems with vector inputs and tensor-structured outputs. The model is motivated by the idea that in this problem the regression regression coefficients are naturally organised in a tensor, and imposing a low-rank structure on such tensor can provide better generalization when the output coefficients exhibit non-trivial correlations related to the underlying tensorial structure. The proposed solution comes with approximation and generalization guarantees, and an empirical evaluation with real multivariate time series data is provided.

Qualitative Assessment

The most interesting aspect of this paper is the efficiency and statistical accuracy of the method demonstrated in the experimental section and backed up by the theory. The theoretical analysis could be more complete, e.g. by including analysis of the running time of the kernel version, an extension of Theorem 3 to the kernel version, and by exploring the possibility of combining Theorem 2 with a two-sided version of Theorem 3 to compare the risk between the learned hypothesis and the optimal hypothesis in the class.

Confidence in this Review

2-Confident (read it all; understood it all reasonably well)


Reviewer 2

Summary

This paper focuses on regression for tensor-form output. A higher-order low-rank regression (HOLRR) algorithm is presented together with its kernel extension. This problem seems quite interesting and the technical contents have some merits. However, the problem is not well motivated from the beginning and only a weak example is presented at the end. The experimental evaluation is incomplete to me. Please see detailed comments below.

Qualitative Assessment

1. The abstract mentioned about tensor-structured output, which is quite interesting to me. However, the Introduction (Sec. 1) did not motivate the problem. It was simply mentioned there is not much work for tensor-structured output data (the end of Par 1 Sec 1) and the authors consider this case for regression (the first sentence of Par 2 Sec 1). However, it is not clear in what applications such problems arise and what are the existing approaches in dealing with such output data. Consequently, the experimental evaluation design is not satisfactory. 2. The end of Sec 1., I expect more details on how the generalization bounds provided are related to the works in [26, 19]. 3. Sec. 2.1, some definitions are not precise and even wrong. E.g., a tensor is not simply a collection of real numbers and the relationships between HOSVD, Tucker decomposition, and higher-order/multilinear PCA seem unclear. 4. The methods need to determine P+1 parameters, R_0 to R_p. Any principled way to determine them? 5. Sec. 4., while the authors point out that MLMLT-NC becomes intractable for large data sets, the real data tested are not large by today's standard. The real data sizes are only 15x125, 15x121, 16x5x5=16x25. 6. Sec. 4: more methods should be included in comparison. In particular, kernel extensions of RLS and LRR should be studied and compared. In other words, kernel LRR should be evaluated as well. Since the authors have kernel HOLRR already, I assume it is not difficult to have kernel LRR. 7. What are the parameter space searched in cross validation for parameters R_0 to R_p in both synthetic and real data experiments? What are those for the kernel parameters in K-HOLRR? 8. Figure 2 or Sec. 4.1., what about the results of Greedy? 9. Table 2 or Sec. 4.2., what about the results of RLS and LRR (we can always vectorize the output and the reshape back)?

Confidence in this Review

3-Expert (read the paper in detail, know the area, quite certain of my opinion)


Reviewer 3

Summary

This paper propose a new algorithm for linear regression with tensor output with theoretical guarantees.

Qualitative Assessment

Regression with tensor response is an important but under-investigated problem. This paper considers linear regression problem and its extension to RKHS regression. The algorithms are intuitive and theoretical analyses are non-trivial. Theorem 3 may be used in other settings as well. It would be interesting to see an algorithm with constant approximation (theorem 2 depends on p). Overall I think this paper should be accepted.

Confidence in this Review

2-Confident (read it all; understood it all reasonably well)


Reviewer 4

Summary

This paper proposes a tensor-variate multilinear regression model called HOLRR for the prediction of a tensor output from vector input. They do this by formulating the problem as the minimization of a least squares subject to a multilinear rank constraint on the regression tensor. A kernelized extension of HOLRR for nonlinear setting is also introduced.

Qualitative Assessment

Strength: --The paper provides the theoretical analysis of approximation guarantees and a generalization bound for the class of tensor-valued regression functions. --The paper is clearly written for readers to understand and appreciate the proposed algorithm. Weakness: --A major drawback is that the novelty and contribution is rather limited. The key idea and the model of this paper is actually equivalent to the HOPLS in the following paper: [Zhao et. al 2013a] "Higher order partial least squares (HOPLS): a generalized multilinear regression method. PAMI". In HOPLS, it assumes the tensor input has low-rank structure and also the tensor output has low-rank structure, and the link of them is established in the common latent space. Specifically, HOPLS projects both input tensor and output tensor simultaneously onto a common latent space by using Tucker models. And then follows a regression step against the projected latent variables. In HOLRR, this equivalence can be seen by taking look at the final regression function f^star in the paragraph "low-rank regression function" of Section 3. For the input vector x: the term x^T*U_0 in f^star is in fact the latent variables, where the factor matrix U_0 projects the input vector x onto a common latent space vector. For the output tensor Y: the authors assume tensor Y has a low-rank Tucker model structure, where G is in fact the core tensor of Y and U_1 to U_p are the loadings. The regression from x to Y is linked by the latent vector as follows: f^star = (x^T*U_0)*G_(1) (U_p kron ... kron U_1), which is essentially the same as HOPLS, the correspondence can be seen clearly in Section 3.4 in the paper [Zhao et. al 2013a]. Thus, the model HOLRR is just a special case of model HOPLS where the input takes form of matrix (stacked vector samples) and the output assume to be a low-rank tensor. The kernelized version is very similar to the Kernel-based tensor PLS regression which can be found in the paper: [Zhao et. al 2013b] "Kernelization of tensor-based models for multiway data analysis: Processing of multidimensional structured data". --This paper only considers the correlations in the output tensor, and hence ignores (destroys) the input-side tensor structure by the vectorization of input data into a vector form, which may suffer the similar limitations as those ignoring the output-side structure. In practice, most real-world tensor-variate regression applications involve tensor covariates input, which may limit the applicability of this model. --For the algorithms, the authors simply apply the truncated HOSVD to find the projection matrices U_p for each mode, and HOSVD is usually used as a initialization point for other algorithms. I doubt that HOSVD is not as accurate as ALS-based HOOI that is used in algorithms such as HOPLS. --For real-data experiments, the HOLRR is mainly tested on the spatio-temporal task against ADMM and GREEDY. The spatio-temporal data is a special case of general tensor data. I think some tasks involving general tonsorial input and output data should be tested. In order to show the superiority of the proposed HOLRR, I think HOPLS [Zhao et. al 2013a] and the corresponding Kernel-based tensor PLS regression [Zhao et. al 2013b] should also be included in comparison on the general data.

Confidence in this Review

3-Expert (read the paper in detail, know the area, quite certain of my opinion)